# Opposite Modulation of the NMDA Receptor by Glycine and S-Ketamine and the Effects on Resting State EEG Gamma Activity: New Insights into the Glutamate Hypothesis of Schizophrenia

**DOI:** 10.3390/ijms24031913

**Published:** 2023-01-18

**Authors:** Moritz Haaf, Stjepan Curic, Jonas Rauh, Saskia Steinmann, Christoph Mulert, Gregor Leicht

**Affiliations:** 1Department of Psychiatry and Psychotherapy, Psychiatry Neuroimaging Branch (PNB), University Medical Center Hamburg-Eppendorf, 20246 Hamburg, Germany; 2Center of Psychiatry, Justus-Liebig University, 35392 Giessen, Germany

**Keywords:** E/I balance, EEG, S-ketamine, glycine, NMDAR, γ oscillations

## Abstract

NMDA-receptor hypofunction is increasingly considered to be an important pathomechanism in schizophrenia. However, to date, it has not been possible to identify patients with relevant NMDA-receptor hypofunction who would respond to glutamatergic treatments. Preclinical models, such as the ketamine model, could help identify biomarkers related to NMDA-receptor function that respond to glutamatergic modulation, for example, via activation of the glycine-binding site. We, therefore, aimed to investigate the effects of opposing modulation of the NMDA receptor on gamma activity (30–100 Hz) at rest, the genesis of which appears to be highly dependent on NMDA receptors. The effects of subanesthetic doses of S-ketamine and pretreatment with glycine on gamma activity at rest were examined in twenty-five healthy male participants using 64-channel electroencephalography. Psychometric scores were assessed using the PANSS and the 5D-ASC. While S-ketamine significantly increased psychometric scores and gamma activity at the scalp and in the source space, pretreatment with glycine did not significantly attenuate any of these effects when controlled for multiple comparisons. Our results question whether increased gamma activity at rest constitutes a suitable biomarker for the target engagement of glutamatergic drugs in the preclinical ketamine model. They might further point to a differential role of NMDA receptors in gamma activity generation.

## 1. Introduction

Although significant advances have been made in understanding the underlying pathomechanisms of schizophrenia over the past decades, this is not reflected in today’s options for pharmacotherapy of the disorder. To date, all approved antipsychotics essentially aim to reduce dopaminergic neurotransmission in the striatum through different mechanisms [1,2]. While these antipsychotics show satisfactory effects on positive symptoms, such as delusions, hallucinations, and thought disorders, in the majority of patients, many patients continue to be affected by treatment resistance and relapse [1,3]. In addition, the benefit for negative symptoms (e.g., avolition, anhedonia, etc.) and cognitive deficits is negligible in most cases [4,5]. These unmet needs are reflected in the high pathological burden of the disease and a deterioration of social and occupational functioning [4], underscoring the need to target alternate neurotransmitter systems [6,7].

Crucial findings have highlighted the glutamate system as a compelling target for the development of new pharmacological treatments [8,9]. In particular, a pathologically relevant hypofunction of the N-methyl-d-aspartate receptor (NMDAR), resulting in an excitation to inhibition (E/I) disbalance [9,10], has been hypothesized based on findings from genetic [11] and post-mortem studies [12]. Several agents aimed at increasing NMDAR function via the activation of the glycine-binding site, and thereby normalizing E/I balance, have shown promising results in preliminary studies. The activation of this allosteric site binding site is essential to enable signal transduction following the engagement of glutamate. Further, glycine promotes and enhances the binding of glutamate to the NMDAR [13]. A direct approach aims to modulate the NMDAR co-agonist binding site through the application of glycine or d-serine [14]. Indirect approaches aim to increase levels of the physiological glycine-binding site agonist d-serine in the synaptic cleft by inhibiting its degradation (i.e., d-amino acid oxidase [DAAO] inhibitors) [15] or the uptake in glia cells (i.e., sodium- and chloride-dependent glycine transporter 1 [GlyT1] inhibitors) [16]. However, most of these agents have failed to demonstrate superior efficacy to placebo in clinical trials, and none have been clinically approved to date.

Despite the overall unsuccessful clinical trials [16,17], reports imply that this treatment approach may benefit a subset of patients or certain symptom domains [9,15]. Therefore, the identification of patients with a relevant E/I disbalance and a glutamate system responsive to NMDAR modulation has become an important goal towards individualized pharmacotherapy of schizophrenia. Preclinical models, such as rodent models or pharmacological disease models in healthy humans, can help identify biomarkers related to E/I balance that respond to modulation of NMDAR function [18,19,20]. Notably, ketamine administration in healthy individuals transiently induces NMDAR hypofunction associated with schizophrenia-like symptoms and electrophysiological alterations [21,22]. Ketamine acts as an uncompetitive antagonist of the NMADR by binding inside the ion pore and thereby blocking signal transduction [18]. This so-called ketamine model of schizophrenia offers the opportunity to investigate the modulation of an exogenically induced E/I disbalance through co-agonists of the NMDAR in the ketamine model of schizophrenia and to identify corresponding neurophysiological alterations.

Deviant fast oscillations in the gamma frequency range (30–100 Hz) of the electroencephalogram (EEG) and magnetoencephalogram (MEG) are a striking neurophysiological alteration, and recent evidence suggests that they may act as markers of E/I imbalance based on impaired NMDAR neurotransmission [19,23,24]. This is of considerable interest since cortical and subcortical Gamma band oscillations (GBOs) play an important part in the coding of information by synchronizing local neuron populations, an essential process in cognition, planning, goal choices, and perception [25,26,27]. Furthermore, the genesis of GBOs crucially involves a motif entailing the excitation of fast-spiking inhibitory GABAergic interneurons via NMDARs. One of the main functions of these interneurons is balancing and synchronizing the excitatory signaling in the brain. In a thoroughly investigated feedback loop, glutamatergic pyramidal cell activity leads to feedback inhibition mediated via the aforementioned interneurons [28].

Resting state activity, recorded during a relaxed wakefulness, is increasingly interpreted as an indicator of intrinsic brain function [29], and elevations of spontaneous gamma oscillations have been repeatedly observed in schizophrenia [23,30,31,32]. However, it remains unclear whether increased gamma oscillations are a hallmark of specific disease stages or might reflect the representation of subgroups with relevant NMDAR deficits, as evidenced by heterogeneous findings in people with schizophrenia (cf. [33]). Comparably, increased gamma activity is a characteristic of altered resting state patterns following the infusion of NMDAR antagonists in healthy individuals [34,35,36] and rodents [37,38], frequently accompanied by a reduction in alpha activity [34,36]. The observed increases in gamma resting state activity are hypothesized to constitute an increased background ‘noise’ [23]. The underlying generators of oscillatory activity at rest are theorized to act as hubs of dynamically organized functional networks that execute distinct tasks [36,39], of which three networks exhibit marked irregularities in schizophrenia [40,41].

The triple network model of schizophrenia comprises the normally anticorrelated default mode network (DMN, i.e., self-referential autobiographical functions) and central executive network (CEN, i.e., attention and processing of exogenous stimuli), as well as the salience network (SN). Converging neuroimaging studies imply that the SN switches between the anticorrelated DMN and CEN by causally affecting the activity of these networks, thereby directing attention to internal or external salient stimuli, respectively [41]. This suggests that a dysfunctional shift between the DMN and CEN, and consequently a reduced anticorrelation, could confound internal and external processing, a hallmark of schizophrenia symptoms [36,42].

Although recent studies have shown that altered stimulus-based gamma activity can be attenuated by pretreatment with NMDAR co-agonists (e.g., glycine) in the NMDAR antagonist model [43,44], to our knowledge, there are no reports on whether these pretreatments can reverse increased resting gamma activity.

Based on the outlined considerations, we aimed to investigate whether the resting state EEG activity in the gamma frequency range could serve as a potential biomarker related to E/I balance that responds to modulation of NMDAR function. To this end, we analyzed the effect of a pretreatment with the NMDAR co-agonist glycine on the ketamine-induced alterations of psychometric measures and resting gamma activity in 25 healthy participants in a placebo, controlled, within-subject design. We hypothesized that pretreatment with glycine (i) attenuates S-ketamine-associated increases of the Positive and Negative Syndrome Scale (PANSS) and 5-Dimensional Altered State of Consciousness (5D-ASC) ratings and (ii) normalizes increased gamma EEG scalp and source activity, as well as functional connectivity, at rest during S-ketamine infusion.

## 2. Results

### 2.1. Participants

We included 25 healthy male participants in the study. Their mean age was 24 years (19–32 years, SD 3.7 years), and they had an average of 16.4 years of education (13–21 years, SD 2.4 years). All participants were right-handed (mean 77.3%, 48–100%, SD 16.4%) and had a mean verbal IQ of 110.9 points (97–122 points, SD 6.8 points) according to the German Wortschatztest (WST) vocabulary test.

### 2.2. Psychometric Assessment

S-ketamine increased PANSS total scores, as well as all PANSS factor scores (Figure 1a), as shown by the significant main effect of the ketamine condition (PANSS total: F(1, 24) = 127, *p* < 0.001, ηp^2^ = 0.84; positive symptoms factor score: F(1, 24) = 34.4, *p* < 0.001, ηp^2^ = 0.59; negative symptoms factor score F(1, 24) = 109, *p* < 0.001, ηp^2^ = 0.82; disorganization symptoms factor score F(1, 24) = 129.1, *p* < 0.001, ηp^2^ = 0.84; emotional distress factor score excitement factor score F(1, 24) = 45.5, *p* < 0.001, ηp^2^ = 0.66; F(1, 24) = 39.5, *p* < 0.001, ηp^2^ = 0.62). In contrast, glycine and the interaction between glycine and ketamine did not significantly affect the PANSS total score or any of the PANSS factor scores.

In the 5D-ASC, we observed a significant interaction between glycine and S-ketamine for the Oceanic Boundlessness dimension (OBN: F(1, 24) = 6.7, *p* = 0.016, ηp^2^ = 0.21). However, this interaction effect did not survive FDR correction. Still, a simple main effects analysis (Bonferroni-corrected) revealed a significantly increased value for the OBN dimension under the ketamine conditions after both placebo (*p* < 0.001) and glycine pretreatment (*p* < 0.001), with OBN values increasing even further when glycine pretreatment preceded ketamine application (*p* = 0.031), while this effect was not evident for placebo-related OBN values (*p* = 0.88). Moreover, S-ketamine significantly increased all dimensional scores (Figure 1b) and total scores (Dread of Ego Dissolution: F(1, 24) = 31, *p*<0.001, ηp^2^ = 0.56; Visionary Restructuralization: F(1, 24) = 40.2, *p* < 0.001, ηp^2^ = 0.63; Auditory Alteration: F(1, 24) = 16.6, *p* < 0.001, ηp^2^ = 0.41; Vigilance Reduction: F(1, 24) = 60.3, *p* < 0.001, ηp^2^ = 0.72; total: F(1, 24) = 60.6 *p* < 0.001, ηp^2^ = 0.72).

### 2.3. EEG Scalp Analysis

We observed differences in oscillatory scalp activity over all four conditions (Figure 2a,b) in three clusters differing regarding the frequency range and involved electrodes: (a) a whole-head cluster encompassing the delta frequency range (*p* = 0.042; 1–4 Hz, Figure 2b,c); (b) a whole-head cluster comprising the alpha to mid beta frequency range (*p* < 0.0001; 8–25 Hz, Figure 2c,d); and (c) a cluster encompassing the central and occipital channels centered around the midline over the whole gamma range (*p* < 0.0001; 30–100 Hz, Figure 2e).

In the follow-up contrasts, S-ketamine significantly altered oscillatory power at the scalp in two different ways: increasing the power of gamma activity (*p* < 0.001; 30–100 Hz, Figure 2h) and reducing the power of slower oscillations (*p* = 0.004; 8–25 Hz, Figure 2g). Ketamine-induced reductions of the power in the delta frequency range (1–4 Hz) did not reach statistical significance (*p* = 0.059, Figure 2f). On the other hand, neither pretreatment with glycine nor the interaction of glycine and S-ketamine significantly altered resting state EEG activity on the scalp.

### 2.4. EEG Source Analysis

In a whole-head analysis, ketamine-induced increases of source activity in the gamma frequency range were restricted to three cortical clusters (all p_FDR_ < 0.05): (a) a large cluster encompassing bilateral medial structures (i.e., dorsal anterior cingulate cortex, orbitofrontal cortex, ventromedial prefrontal cortex, and retrosplenial cortex), as well as the right dorsolateral prefrontal cortex and right temporal areas; (b) a cluster in the left visual cortex; and (c) a cluster enclosing the left angular gyrus and associative visual cortex (Figure 3b). Notably, the largest effect sizes were found for voxels within the bilateral anterior cingulate cortex (ACC, Figure 3a). On the other hand, neither pretreatment with glycine nor the interaction of glycine and S-ketamine significantly altered whole-head resting state EEG source activity.

Using ROI-based analysis to assess gamma source activity, we found an interaction effect between glycine pretreatment and S-ketamine on the mean source power of CEN (F(1, 24) = 4.5, *p* = 0.043, ηp^2^ = 0.16, Figure 3c) and DMN ROIs (F(1, 24) = 6.1, *p* = 0.02, ηp^2^ = 0.20, Figure 3d), but neither survived FDR correction. However, S-ketamine significantly increased the mean source activity of the DMN (F(1, 24) = 9, *p* = 0.006, ηp^2^ = 0.27, Figure 3d) and the SN ROIs (F(1, 24) = 19, *p* = 0.0002, ηp^2^ = 0.44, Figure 3e).

### 2.5. EEG Functional Connectivity Analysis

To assess ketamine-induced alterations in gamma-band functional connectivity, we analyzed the changes in lagged coherence within each of the three resting state networks (CEN, DMN, and SN, Figure 4a–c) and between each network and the other two networks (Figure 4d–f). As for intra-network connectivity, S-ketamine significantly increased the mean gamma-band functional connectivity between all ROIs located within the SN (F(1, 24) = 6.2, *p* = 0.02, ηp^2^ = 0.21, Figure 4c), whereas the connectivity within the DMN, as well as within the CEN, was not affected. However, after FDR correction, this change in gamma connectivity within the SN did not remain significant.

As for inter-network connectivity in the gamma frequency range, S-ketamine increased the mean connectivity between ROIs of the CEN and the DMN ROIs (F(1, 24) = 5.3, *p* = 0.03, ηp^2^ = 0.18, Figure 4d) and between ROIs of the CEN and the SN ROIs (F(1 ,24) = 5.7, *p* = 0.026, ηp^2^ = 0.19, Figure 4e), as well as between ROIs of the DMN and the SN ROIs (F(1, 24) = 5.9, *p* = 0.023, ηp^2^ = 0.20, Figure 4f). These connectivity changes did not remain significant after FDR correction.

Neither glycine nor the interaction of glycine and S-ketamine significantly altered resting state functional connectivity in the gamma band.

### 2.6. Association between the Psychometric Results and Neurophysiological Measures

To examine the association between psychometric scores and neurophysiological measures, we used a multiple linear regression model including all five PANSS factor scores to predict ketamine-induced changes in the significantly affected neurophysiological measures. This regression model predicted the mean scalp gamma activity (F(5, 20) = 6.2, *p* = 0.001) with an *R^2^* of 0.62. On the one hand, increases in the Positive (b = 0.0021 CI 0.0002–0.004) and Disorganization factor scores (b = 0.0022 CI 0.0007–0.0037) corresponded to increases of the scalp gamma activity, both following the administration of S-ketamine, irrespective of pretreatment. On the other hand, increases in Emotional Distress parallel reductions of the scalp gamma activity (b = −0.0052 CI −0.0079–−0.0025). None of the changes in CEN, DMN, or SN source activity could be predicted by this regression model.

A second regression model that included all dimensional values of the 5D-ASC questionnaire did not significantly predict scalp gamma activity (F(5, 20) = 2.7, *p* = 0.054). However, the Oceanic Boundlessness (r = 0.46, *p* = 0.022) and the Visionary Restructuralization dimension scores (r = 0.47, *p* = 0.013) were both positively correlated with the mean gamma scalp activity.

## 3. Discussion

In this study, we investigated whether a pretreatment with the NMDAR co-agonist glycine would normalize ketamine-induced alterations of resting state EEG activity in the gamma frequency range (30–100 Hz) and changes in psychopathological ratings. As expected, S-ketamine engendered schizophrenia-like symptoms and an altered state of consciousness, accompanied by increased scalp and source gamma activity, as well as functional connectivity, in the healthy participants. The emerging schizophrenia-like symptomatology predicted changes in gamma scalp activity, which was also positively associated with the Oceanic Boundlessness and Visionary Restructuralization dimension scores. While pretreatment with glycine appeared to attenuate ketamine-induced increases of gamma source activity in the CEN and DMN ROIs and seemed to increase the Oceanic Boundlessness dimension score when combined with S-ketamine, none of these glycine effects remained significant after correction for multiple testing. Thus, contrary to our hypotheses, we were unable to demonstrate significant effects of glycine on ketamine-induced changes of gamma activity at rest.

The increase in resting state EEG activity observed in this study after S-ketamine administration corroborates the consistent findings of increased gamma scalp [35,36,45,46] and source activity [36,47], as well as functional connectivity [45,48], in human participants. A comparable increase in localized and generalized gamma activity can be induced in rodents by administration of NMDAR antagonists (e.g., ketamine, PCP, MK−801), although a minority of studies found this effect to be dose-dependent (see for review [30]). In the studies reporting dose-dependent effects, low doses—comparable to subanesthetic doses in humans—generally increased gamma activity, whereas high doses tended to reduce it. Despite the recurrent patterns in resting state gamma activity caused by exogenous blockade of the NMDAR, heterogeneous findings have been reported in individuals with schizophrenia (see for review [30]). Increases [32,49], decreases [50], as well as no changes [51,52,53], in resting state gamma activity have been reported. Various explanations are conceivable for these heterogeneous observations in people with schizophrenia: it is possible that only certain subgroups show significant increases in gamma activity at rest, and their representation in the study sample determines the overall results, or that these increases are a hallmark of certain disease stages and reflect stage-specific pathophysiological changes.

These changes in gamma activity at rest are generally attributed to blockage or hypofunction of the NMDAR on parvalbumin-expressing (PV+) GABAergic interneurons, which are thought to be critically involved in local feedback inhibitory circuits that encompass glutamatergic pyramidal cells [28]. Reciprocal excitation and inhibition within these local feedback inhibitory circuits is thought to underlie the generation of gamma rhythms, which have therefore been proposed to reflect E/I balance [28]. In this framework, ketamine-induced hypofunction of the NMDAR is hypothesized to dysregulate feedback inhibition mediated by PV+ interneurons, resulting in disinhibited and excessive glutamate release from pyramidal cells associated with changes in gamma oscillations. Notably, people with schizophrenia exhibited morphological alterations of dendrites and a loss of interconnected PV+ interneurons according to postmortem studies [12,54]. However, a recent study compared the effect of NMDAR knock-out in PV+ interneurons to the administration of ketamine, and while it found marked increases in gamma activity for both samples, the activity patterns in both groups differed substantially [55]. This finding suggests that diverging mechanisms, instead of or in addition to NMDAR blockage on PV+ interneurons, may be responsible for the broadband increase in resting state gamma activity after ketamine administration and raises the question of whether the increase in activity reflects oscillatory activity [55]. Our results support the assumption of diverging, non-NMDAR-dependent mechanisms, as the NMDAR co-agonist glycine did not affect the ketamine-associated increase in gamma activity at rest, especially considering that the only receptor affected by both drugs is the NMDAR, and our previous results demonstrated that glycine markedly attenuated the ketamine-induced reduction in stimulus-evoked gamma oscillations [43].

In our study, the changes in gamma activity in the surface EEG emanating from S-ketamine administration were reflected in the altered source activity of nodes in the three-network (TN) model, resembling previously reported changes [36], a major exception being the CEN, which did not display increases of gamma activity in our study. An imbalance between the CEN (involved in external processing) and the DMN (involved in internal processing), which are generally anticorrelated networks, is thought to cause several core symptoms of schizophrenia, such as delusions or thought insertion and withdrawal [56,57]. In particular, the inability to simultaneously increase CEN activity and decrease DMN activity is a hallmark of the disease [56,57,58]. This inability appears to be accompanied by a reduction in the modulatory activity of the SN [59] and especially the anterior insula [60], resulting in a disbalance of the immanent inverse activity. This may be reflected in our observations of increased source activity in the DMN, but not the CEN, after S-ketamine administration. Against this interpretation is the fact that we found a close relationship between gamma activity at the scalp level, but neither with source activity nor with functional connectivity. Significant increases in gamma functional connectivity have also been observed in patients with schizophrenia [61,62], as well as healthy humans, following the administration of ketamine [45,48]. Yet, the comparability of these results is limited by differing methodological approaches.

Remarkably, gamma activity in people with schizophrenia and in NMDAR-antagonistic disease models displays opposite changes in stimulus-based recordings than in resting state recordings: stimulus-evoked gamma oscillations demonstrate marked reductions in persons with schizophrenia across all stages of the disease [63,64,65,66,67], as well as in healthy humans [24,43] and rodents [37,44], following the infusion of NMDAR antagonists. While gamma activity under the influence of NMDAR antagonists increases over a wide range from 30 to 100 Hz (and presumably beyond) at rest, evoked gamma reductions center around 40 Hz (cf. [43,63,65]). This could be another indication of distinct phenomena in the generation of gamma activity, in which the NMDAR plays a differentially important role, and it is also supported by the observation that glycine pretreatment attenuates ketamine-induced changes of stimulus-evoked gamma oscillations (where the effect of pharmacological modulation is concentrated around 40 Hz; cf. [43]), but not resting state.

To our knowledge this is the first study to investigate the modulation of resting state EEG via glutamatergic agents in the ketamine model of schizophrenia [21]. This fact enables only indirect comparisons with other neuroimaging modalities. A recent resting state study investigated the impact of two metabotropic glutamate receptor (mGluR) 2/3 agonists on ketamine-induced activity increases in the dACC employing fMRI: Pomaglumetad had no effect on these region-specific activity increases, whereas TS-134 attenuated the increases, but only at the lower of the two doses [68]. Correspondingly, agonists of mGluR2 and mGluR2/3 at the highest doses reduced activity increases after ketamine administration in several predefined ROIs [69]. However, these observations probably cannot be attributed to the modulation of NMDAR-dependent neurotransmission, but probably reflect the attenuation of a consecutive excessive glutamate release [68]. While several studies have reported that NMDAR co-agonists (e.g., glycine or d-cycloserine) can normalize event-related potentials in people with schizophrenia [70,71], no observations on the effects of glutamatergic modulation on resting state activity have been published.

Finally, several limitations concerning our results merit consideration. While our small sample size and homogeneous study population limit generalizability, the absence of any glycine effect on scalp EEG activity suggests at most a subtle modulation, if any, of the resting state gamma activity changes associated with S-ketamine. Nevertheless, some interaction effects concerning source activity with large effect sizes were observed, but these no longer remained significant after correction for multiple comparisons, making it difficult to distinguish whether they are true or false negatives, a question that can only be addressed with a larger sample size. While the only receptor directly affected by glycine and S-ketamine is the NMDAR, the downstream interplay of any further modulated neurotransmitter systems could represent an indirect, non-opposing interaction between the two drugs. However, the lack of significant interaction effects and the absence of glycine effects render a relevant downstream interaction unlikely. Still, methodological limitations of EEG measurements limit the information that can be obtained about the interaction of these factors at the receptor level. Finally, the concept of a fixed, as opposed to a dynamic, resting state most likely does not reflect the complexity of this uncontrolled brain state, which involves various cognitive and perceptual processes over the duration of the experiment.

In conclusion, we found that gamma activity at rest did not respond to enhancement of NMDAR function in the ketamine model of schizophrenia. Therefore, our results question whether increased gamma activity at rest constitutes a suitable biomarker related to E/I balance for the target engagement of glutamatergic drugs in the preclinical ketamine model. While the ketamine model of schizophrenia resembles the disease in several qualities, it remains to be proven whether our results extend to people with schizophrenia, calling for the investigation of the effect of glutamatergic agents on the gamma resting state activity in schizophrenia. Finally, the possible differential role of the NMDAR in gamma activity generation highlighted in this work needs to be investigated in more detail.

## 4. Materials and Methods

### 4.1. Participants

Only male participants without a history of neurological or psychiatric disease were included in this study. The general procedure was approved by the Ethics Committee of the Hamburg Medical Association (PV4575—01.11.2016) and was performed in accordance with the latest version of the Declaration of Helsinki. Written informed consent was obtained from all participants after they had been informed in detail about the nature of the procedures. One participant discontinued the study because of adverse effects of S-ketamine (dissociative effect/headache). Participants were recruited from the general population by advertising or word of mouth, and they were either medical students or medical staff of the University Hospital Hamburg or students at the University of Hamburg.

Inclusion criteria were male sex, age between 18 and 40 years, right-handedness, normal or corrected-to-normal visual acuity, German language at native level, and normal hearing. The sample was limited to participants of male sex based on safety concerns regarding the application of S-ketamine in case of an undetected pregnancy. Exclusion criteria were illicit drug use, acute or past psychiatric disorders (assessed with the Mini International Neuropsychiatric Interview [72] and the Schizotypal Personality Questionnaire (SPQ) [73]) or treatment, family history of schizophrenia or bipolar disorder, neurological disorders, cardiovascular disease, thyroid disease, current medications, and ketamine or glycine intolerance.

Participants received financial compensation of 180 euros for participating in the study.

### 4.2. Study Design

The study was conducted in a randomized, placebo-controlled, crossover design. Participants were blinded to pretreatment and continuous infusion, whereas the clinical evaluator was blinded only to pretreatment.

All participants underwent four test sessions, during which participants were pretreated with either glycine or placebo and then received a continuous infusion of S-ketamine or placebo. This resulted in four different experimental conditions: (1) S-ketamine with glycine pretreatment (GlyKet), (2) S-ketamine with placebo pretreatment (PlaKet), (3) placebo with glycine pretreatment (GlyPla), and (4) placebo with placebo pretreatment (PlaPla). Electroencephalography (EEG) recordings were performed on participants during continuous infusion in all four sessions. The order of sessions was randomized and overall counterbalanced, and there was a minimum interval of five days between each of the four test sessions.

To reduce the risk of nausea and vomiting and to avoid potentiating interactions, participants were required to abstain from food or alcohol, respectively, for at least 12 h before each test session.

Glycine pretreatment was administered at a dose of 200 mg/kg body weight as an intravenous infusion in 500 mL of 0.9% sodium chloride (NaCl) solution over 1 h [70,74]. Placebo was administered analogously as a 500 mL 0.9% NaCl infusion. Both pretreatments were prepared by an unblinded third person in a separate room immediately before the recording sessions. The ready-to-use pretreatment infusions were indistinguishable from each other. During the ketamine sessions, a subanesthetic dose of S-ketamine hydrochloride (Ketanest^®^ S, Pfizer Pharma GmbH, Berlin, Germany) in 0.9% NaCl solution was administered intravenously using a syringe pump (Perfusor^®^Space, B. Braun, Melsungen, Germany) for a total duration of 75 min. The S-ketamine infusion started with an initial bolus of 10 mg over 5 min, followed by a maintenance infusion of 0.006 mg/kg/min [24,75]. Because plasma ketamine levels increase slowly with continuous infusion [76], the dose was reduced by 10% every 10 min, in accordance with previously published protocols [77]. Placebo was administered analogously as a 0.9% NaCl infusion.

Heart rate, blood pressure, and oxygen saturation were monitored continuously during all sessions. In addition, participants were monitored via webcam to ensure their safety and compliance with paradigm instructions. After the sessions, participants remained under constant supervision for one hour until all effects of the drug had worn off. They were then released to the care of a friend or relative. Participants were not allowed to drive motor vehicles for the next 24 h. The clinical rater was blinded to pretreatment, but not to the continuous infusion, because the clinical effects of S-ketamine were evident.

### 4.3. Psychometric Assessment

Psychiatric symptomatology was assessed using the Positive and Negative Syndrome Scale (PANSS) clinical interview [78]. After each admission session, the interview was always conducted by the same experienced psychiatrist, who was blinded to pretreatment. This served to assess schizophrenia-like symptoms during the admission session.

The PANSS consists of 30 items that can be divided into three domains, including a positive symptoms domain (7 items), a negative symptoms domain (7 items), and a general pathology domain (16 items). As in our previous studies [24,43,79,80], PANSS scores were evaluated using the five-factor model of van der Gaag et al. [81]. This factor model includes the factors (i) positive symptoms, (ii) negative symptoms, (iii) disorganization symptoms, (iv) excitement, and (v) emotional distress, and it is the only model with a satisfactory goodness-of-fit [81].

The 5-Dimensional Altered State of Consciousness (5D-ASC) questionnaire [82,83], a visual analog scale, was administered after each recording session to assess the subjective effects of S-ketamine. This self-assessment questionnaire consists of 94 items assessing 5 key dimensions (subscales) of altered states of consciousness. The Oceanic Boundlessness (OBN) dimension summarizes depersonalization and derealization associated with positive emotional states, ranging from elevated mood to euphoric elation, also described as mystical experiences. The Dread of Ego Dissolution (DED) dimension measures negatively perceived ego disintegration and loss of self-control associated with anxiety, with high scores indicating a very unpleasant state. The Visionary Restructuralization subscale (VRS) includes (pseudo) hallucinations, visions, illusions, and synesthesia. In addition, this dimension includes a change in the perceived meaning of objects. Together, the three factor scores represent primary, etiology-independent dimensions of altered states of consciousness. The two etiology-dependent, and thus secondary, dimensions are Auditory Alteration (AUA) and Vigilance Reduction (VIR) [82].

### 4.4. EEG Aquisition

The recording took place in a sound-attenuated and electrically shielded room. Participants were seated in a chair with a headrest. The EEG was recorded at a sampling rate of 1000 Hz and with an analog bandpass filter (0.1–1000 Hz) with 64 active electrodes, using Brain Vision Recorder software version 1.21 (Brain Products, Gilching, Germany). The electrodes were mounted on an elastic cap (ActiCap, Brain Products , Gilching, Germany) and positioned in an extended 10/20 system, with the additional positions AF7, AF3, AF4, AF8, F5, F1, F2, F6, F10, FT9, FT7, FC3, FC4, FT8, FT10, C5, C1, C2, C6, TP7, CPz, TP8, P5, P1, P2, P6, PO3, POz, and PO4. Eye movements were recorded via four electrooculography channels (bilateral at the outer canthi and right infraorbital and supraorbital). An electrode at the FCz position was used as a reference, and the electrode at the AFz position served as a ground. SuperVisc electrode gel (EASYCAP GmbH, Herrsching, Germany) was applied to establish contact between the scalp and the electrodes. Impedances were always kept below 5 kΩ.

Resting electroencephalographic activity was recorded for five-and-a-half minutes, in accordance with recommended pharmaco-EEG standards, while participants sat in a reclined position with eyes closed [84]. To ensure that participants were comfortable yet awake, they were instructed to open their eyes (after 180 s) and close them (2 s thereafter) when prompted.

### 4.5. EEG Preprocessing

EEG data preprocessing was conducted in Matlab (MathWorks^®^) using the open-source toolbox EEGLAB [85]. To limit the analysis to resting state EEG with eyes closed, segments in which participants had their eyes open (the period between being instructed to first open and then close their eyes and the period after the paradigm ended) were discarded. After filtering the EEG data with a low-pass filter (100 Hz, cutoff frequency 112.5 Hz), as well as a high-pass filter (1 Hz, cutoff frequency 0.5 Hz), sinusoidal line noise was removed from the scalp channels using the CleanLine plugin (line noise frequencies 50 Hz and harmonics, window size 4 s, window step 1 s). Using the Clean Rawdata plugin, the prominent channel, muscle, and movement artifacts (burst criterion: 20 SD, all other parameters were set to default), as well as flat channels (>5 s), were automatically detected and either restored or otherwise rejected from further analysis using Artifact Subspace Reconstruction (ASR) [86]. Following re-referencing to the common average, independent component analysis was performed based on an infomax extended ICA algorithm with PCA dimension reduction for interpolated electrodes (stop criterion: weight change 10^−7^). Components that were marked as neither “Brain” nor “Other” as most likely class with the ICLabel plugin [87] were automatically rejected, while all components marked as “Brain” with a likelihood of less than 50% or as “Other” were subsequently inspected manually. For the interpolation of removed channels, a 4th-order spherical spline interpolation [88] using a regularization parameter lambda of 1 × 10^−5^ was employed.

### 4.6. EEG Scalp Analysis

EEG data analysis was conducted using the open-source toolbox Fieldtrip [89]. The continuous EEG data were segmented into 2 s epochs and subjected to a fast Fourier transformation (FFT) using Hanning windows and a frequency resolution of 0.5 Hz for all frequencies between 1 and 100 Hz. This was followed by the computation of the average resting state activity for each condition and participant. To compare these averages between all four conditions, a non-parametric permutation-based inference was applied to provide a Monte Carlo approximation of the randomization distribution of the dependent samples’ F-statistics involving spatiotemporal clustering for multiple comparisons correction, as implemented in FieldTrip (tail = 1, clusteralpha = 0.05, alpha = 0.05, 100,000 randomizations). Subsequent contrasts for both conditions (i.e., glycine and ketamine), as well as their interaction, were analyzed using dependent samples t-statistics with adopted parameters (tail = 0, clusteralpha = 0.05, alpha = 0.05, correcttail = prob, 100,000 randomizations).

### 4.7. EEG Source Analysis

The scalp data (2 s epochs) were subjected to a frequency analysis to extract the cross-spectra within the gamma frequency range previously identified as significantly different between the four conditions (30–100 Hz). The exact low-resolution brain electromagnetic tomography algorithm (eLORETA [90]) (as implemented in the METH toolbox) was applied to the mean values across all epochs of these complex data for each participant to calculate the source power values for each voxel in the source grid (2839 voxels on the cortical surface). Whole-head comparison between groups was performed, computing voxel-wise statistical tests.

An ROI-based analysis of source activity was performed by extracting the power of the centroid voxel of each of the bilateral Brodmann Areas (BA) for the three networks included in the TN model (i.e., DMN, CEN, and SN) and included in BA 9 (DLPFC-CEN), BA 40 (PPC-CEN), BA 11 (VMPFC-DMN), BA 23/30 (PCC-DMN), BA 24/32 (ACC-SN), and BA 47 (AI-SN). The predefined ROIs were based on the BA definitions provided by the LORETA software package, which is based on the Talairach Daemon. For statistical analysis, results were averaged across all bilateral ROIs of each network for each participant in all four sessions.

### 4.8. EEG Functional Conncectivity Analysis

The gamma-band cross-spectra extracted for EEG source analysis were subjected to functional connectivity analysis in the source space, employing the lagged coherence measure (LC [91]) using the METH toolbox implementation, and were averaged across all epochs. The analysis of functional connectivity focused on connectivity between the ROIs of the TN model within and between networks. For statistical analysis, results were averaged across the LC values between each ROI and all other ROIs within the network (i.e., intra-network connectivity) or all ROIs within another of the three networks (i.e., inter-network connectivity) for each participant in all four sessions.

### 4.9. Statistical Analyses

All statistical analyses were performed in Matlab (MathWorks^®^). Psychometric scores, voxel-wise whole-head source activity, ROI-based source activity, and ROI-based functional connectivity were tested using 2 × 2 repeated measure ANOVA with glycine and ketamine as within-subject factors. The *p*-values obtained by multiple testing were subjected to FDR correction (0.05).

## Figures and Tables

**Figure 1 ijms-24-01913-f001:**
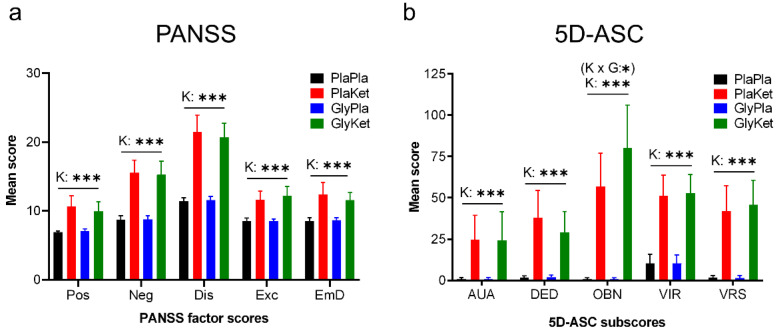
Bar charts of the means (+95% CI) of (**a**) the five Positive and Negative Syndrome Scale (PANSS) factor scores and (**b**) the five 5-Dimensional Altered State of Consciousness (5D-ASC) dimensional scores (K × G: ketamine × glycine interaction term; K: ketamine main effect; * *p* < 0.05; *** *p* < 0.001; effects that were no longer significant after FDR correction are shown in parentheses; Pos: positive symptoms; Neg: negative symptoms; Dis: disorganization symptoms; Exc: Excitement; EmD: emotional distress; AUA: Auditory Alteration; DED: Dread of Ego Dissolution; OBN: Oceanic Boundlessness; VIR: Vigilance Reduction; VRS: Visionary Restructuralization).

**Figure 2 ijms-24-01913-f002:**
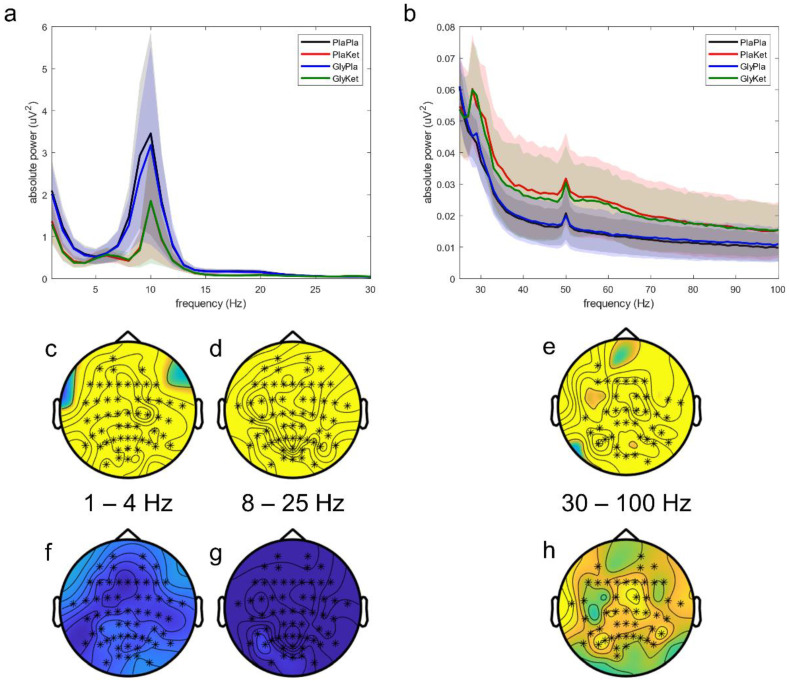
Mean resting state EEG power spectra across all electrodes in (**a**) the frequency range 1–30 Hz and (**b**) the frequency range 25–100 Hz for all four conditions: the PlaPla (black line), PlaKet (red line), GlyPla (blue line), and GlyKet condition (green line). The shaded areas depict the corresponding standard deviation of each condition. Topographical representation of statistical differences among all four conditions (significantly different electrodes marked as asterisks) found in three clusters: (**c**) 1–4 Hz; (**d**) 8–25 Hz; (**e**) 30–100 Hz. Corresponding main effect analyses for ketamine reveal activity decreases in the 1–4 Hz (**f**, not significant) and the 8–25 Hz (**g**) clusters, but activity increases in the 30–100 Hz cluster (**h**) (significantly different electrodes marked as asterisks).

**Figure 3 ijms-24-01913-f003:**
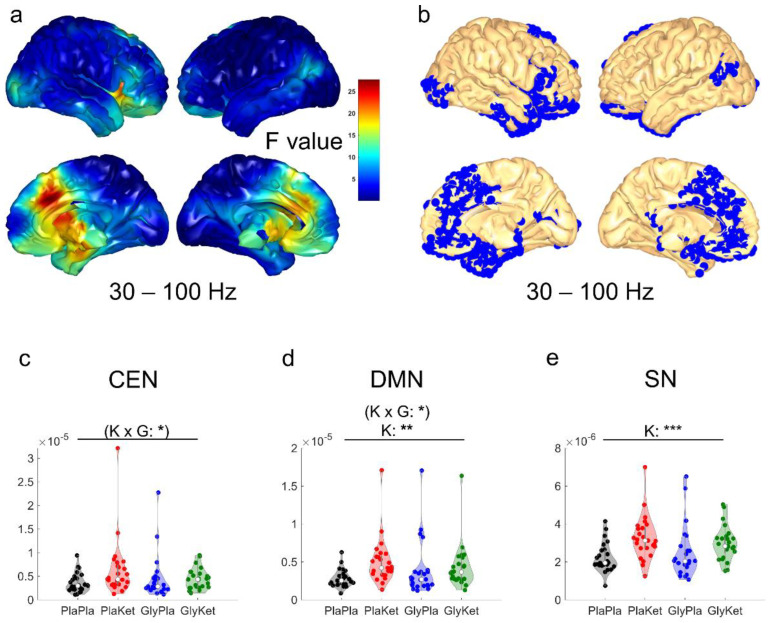
(**a**) Cortical map with F-values obtained by voxel-wise main effect analyses for ketamine of source activity computed as exact low-resolution brain electromagnetic tomography (eLORETA) in the gamma frequency band (30–100 Hz). (**b**) Blue voxels depict corresponding significantly increased (pFDR < 0.05) source activity during the administration of ketamine in three large clusters. Violin plots of the mean network source-localized gamma (30–100 Hz) power of all regions of interest (ROI) in the central executive network (CEN); (**c**), the default mode network (DMN); (**d**), and the salience network (SN); (**e**) for all four conditions: the PlaPla (black dots), PlaKet (red dots), GlyPla (blue dots), and GlyKet condition (green dots). (K × G: ketamine × glycine interaction term; K: ketamine main effect; * *p* < 0.05; ** *p* < 0.01, *** *p* < 0.001; effects that were no longer significant after FDR correction are shown in parentheses).

**Figure 4 ijms-24-01913-f004:**
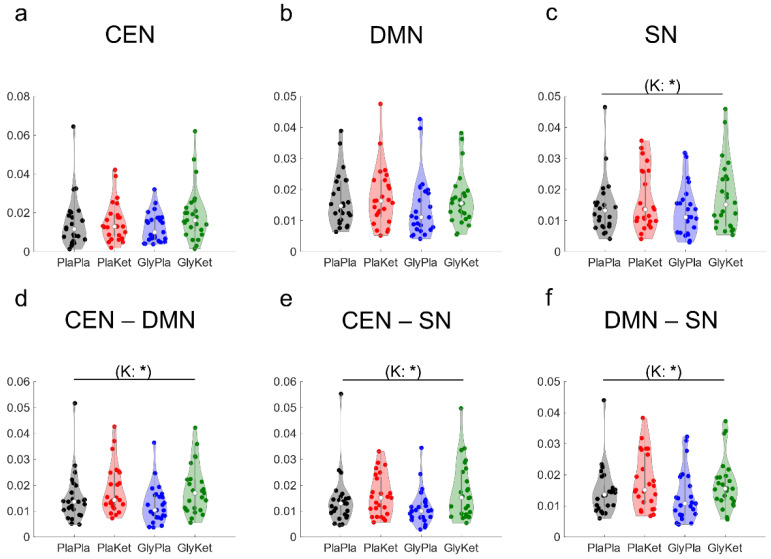
Violin plots of the mean intra-network gamma (30–100 Hz) functional connectivity (computed as lagged coherence) among all regions of interest (ROI) in the central executive network (CEN); (**a**), the default mode network (DMN); (**b**), and the salience network (SN); (**c**) for all four conditions: the PlaPla (black dots), PlaKet (red dots), GlyPla (blue dots), and GlyKet condition (green dots). Inter-network connectivity is displayed as mean lagged coherence between all ROIs of the CEN and the DMN (**d**), between all ROIs of the CEN and the SN (**e**), and between all ROIs of the DMN and the SN (**f**). (K: ketamine main effect; * *p* < 0.05; effects that were no longer significant after FDR correction are shown in parentheses).

## Data Availability

The data presented in this study are available on request from the corresponding author. The data are not publicly available due to privacy restrictions and require approval of a data-sharing agreement.

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
