# Peer review of "Opposite Modulation of the NMDA Receptor by Glycine and S-Ketamine and the Effects on Resting State EEG Gamma Activity: New Insights into the Glutamate Hypothesis of Schizophrenia"

_ijms, 2023, doi:10.3390/ijms24031913_

Round 1

Reviewer 1 Report

Haaf and collaborators looked at the effects of Ketamine (or placebo) treatment following pre-treatment with glycine (or placebo) on PANSS and 5D-ASC scores, and the gamma band activity of healthy male controls. Ketamine induced schizophrenia-like symptoms along with increased gamma activity and functional connectivity. Glycine did not prevent these effects.

Important study well-designed and interesting conclusions.

Introduction:

  • Generally very well written.
  • "Several agents aimed at increasing NMDAR function and thereby normalizing the E/I-balance, however, failed in clinical trials," How did they fail? Because binding works only when clozapine is present? No effect on symptoms? It is not clear. Also this sentence is probably too long, and the paragraph too.
  • "The triple network model [...]" Please initiate a new paragraph here.
  • "the SN switches between the anticorrelated DMN and CEN" This is a fairly complex theory and might need to be explained in other words.

Material and methods:

  • Please clarify why only males were included.
  • "German at native level" please edit to read: "German language at native level"
  • "The study was conducted in a single-blind" Then, two paragraphs later, the reader learns that the pre-treatment condition was double-blinded. Perhaps be more specific within the first paragraph.
  • Please clarify why participant had to abstain from food for 12 hours.
  • It is probably best to present the four experimental condition and then specify that an EEG was performed after each.
  • Methods are otherwise well described.

Results:

  • "As for the PANSS" Implies that other results were presented before, but this is the first one.
  • It could be helpful to see the significant correlations on graphs.
  • The results are otherwise clear and well presented.

Discussion:

  • P.8, line 271: "In this subset" Which subset?
  • "which was accompanied by increased inter-network connectivity between the three resting-state networks examined, although it should be noted that the last observation did not survive correction for multiple comparisons." If the increase is not significant, please do not discuss these results.
  • P.9, line 340: Please cut this first sentence in two.
  • Please describe what Pomaglumetad and TS-134 are.

Overarching comment:

  • It is recommended to use the term 'participant', which reflects the active intention of the person, rather than 'subject', which may be perceived as the person being passively subjected to treatments and measures.

Reviewer 2 Report

This MS explores the role of role of glutamatergic mechanisms in schizophrenia, an important topic since current medication solely relies on targeting the dopaminergic system. The authors are studying the role of a co-agonist of the NMDA receptors (NMDA hypofunction is a characteristic of the dissease) in the ketamine model of schizophrenia. To do so the authors investigated whether a pretreatment with glycine would revert the ketamine-induced increase gamma oscillations and changes in psychopathological ratings. The glycine pretreatment seemed to attenuate ketamine-induced increases of gamma activity and pathological behavior when combined with S-ketamine, but the glycine effects were not significant after correction for multiple testing.

The ms relies on a clear-cut hypothesis, ie that a co-agonist of NMDARs can protect some functions caused by NMDAR hypofunction. The main limitation of the study is it relies on a negative result. It is an important finding for a specializes audience. The MS is well written, figures informative, stats appropriate.

Some comments below:

1.        In the abstract: it is not easy to understand the rationale of pre-treatment with glycine, perhaps

2.       The factors influencing the interactions of the co-agonist and antagonist could be discussed from a pharmacological point of view

3.       How does the co-agonist affect various consequences of GLU hypofunction if caused by ketamine vs. other factors.
